# Pervasive Label Errors in Test Sets Destabilize Machine Learning Benchmarks

**Curtis G. Northcutt**[*]
ChipBrain, MIT, Cleanlab

**Anish Athalye**
MIT, Cleanlab

**Jonas Mueller**
AWS

## Abstract

We identify label errors in the *test* sets of 10 of the most commonly-used computer vision, natural language, and audio datasets, and subsequently study the potential for these label errors to affect benchmark results. Errors in test sets are numerous and widespread: we estimate an average of at least 3.3% errors across the 10 datasets, where for example label errors comprise at least 6% of the ImageNet validation set. Putative label errors are identified using confident learning algorithms and then human-validated via crowdsourcing (51% of the algorithmically-flagged candidates are indeed erroneously labeled, on average across the datasets). Traditionally, machine learning practitioners choose which model to deploy based on test accuracy — our findings advise caution here, proposing that judging models over correctly labeled test sets may be more useful, especially for noisy real-world datasets. Surprisingly, we find that lower capacity models may be practically more useful than higher capacity models in real-world datasets with high proportions of erroneously labeled data. For example, on ImageNet with corrected labels: ResNet-18 outperforms ResNet-50 if the prevalence of originally mislabeled test examples increases by just 6%. On CIFAR-10 with corrected labels: VGG-11 outperforms VGG-19 if the prevalence of originally mislabeled test examples increases by just 5%. Test set errors across the 10 datasets can be viewed at `https://labelerrors.com` and all label errors can be reproduced by `https://github.com/cleanlab/label-errors`.

## 1 Introduction

Large labeled datasets have been critical to the success of supervised machine learning across the board in domains such as image classification, sentiment analysis, and audio classification. Yet, the processes used to construct datasets often involve some degree of automatic labeling or crowd-sourcing, techniques which are inherently error-prone [39]. Even with controls for error correction [20, 49], errors can slip through. Prior work has considered the consequences of noisy labels, usually in the context of *learning* with noisy labels, and usually focused on noise in the *train* set. Some past research has concluded that label noise is not a major concern, because of techniques to learn with noisy labels [31, 35], and also because deep learning is believed to be naturally robust to label noise [17, 28, 38, 43].

However, label errors in *test* sets are less-studied and have a different set of potential consequences. Whereas *train* set labels in a small number of machine learning datasets, e.g. in the ImageNet dataset, are well-known to contain errors [16, 33, 40], labeled data in *test* sets is often considered "correct" as long as it is drawn from the same distribution as the train set. This is a fallacy: machine learning *test* sets can, and do, contain errors, and these errors can destabilize ML benchmarks.

---

[*]Correspondence to: `curtis@cleanlab.ai` or `cgn@csail.mit.edu`.

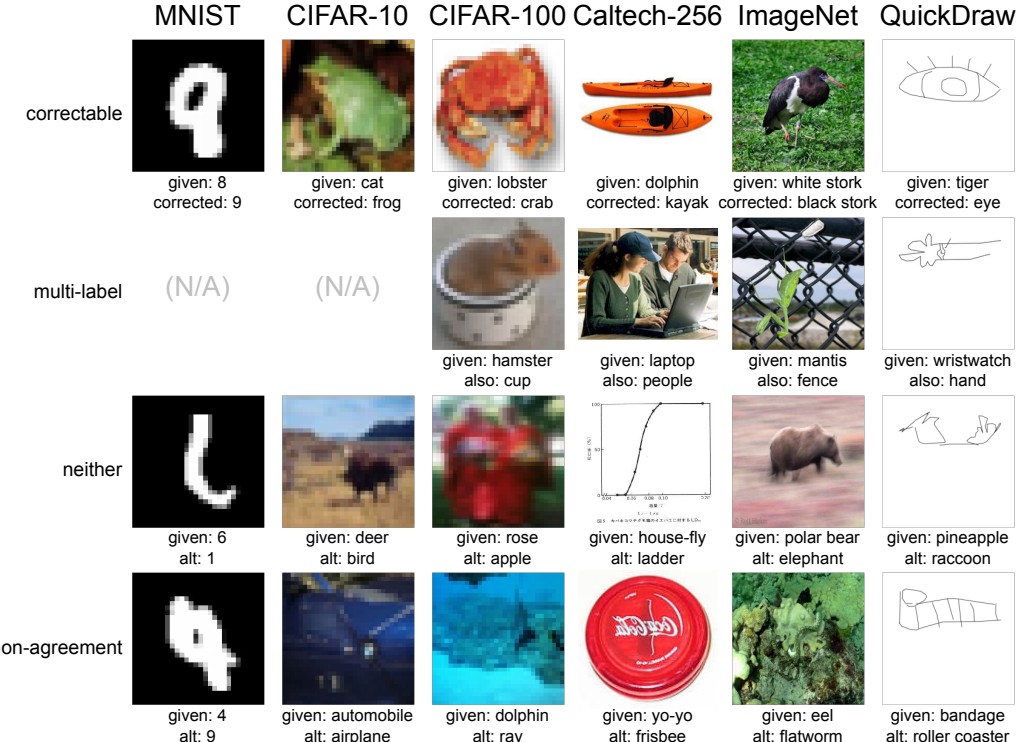

Figure 1: An example label error from each category (Section 4) for image datasets. The figure shows given labels, human-validated corrected labels, also the second label for multi-class data points, and CL-guessed alternatives. A gallery of label errors across all 10 datasets, including text and audio datasets, is available at https://labelerrors.com.

Researchers rely on benchmark test datasets to evaluate and measure progress in the state-of-the-art and to validate theoretical findings. If label errors occurred profusely, they could potentially undermine the framework by which we measure progress in machine learning. Practitioners rely on their own real-world datasets which are often more noisy than carefully-curated benchmark datasets. Label errors in these test sets could potentially lead practitioners to incorrect conclusions about which models actually perform best in the real world.

We present the first study that systematically characterizes label errors across 10 datasets commonly used for benchmarking models in computer vision, natural language processing, and audio processing. Unlike prior work on noisy labels, we do not experiment with synthetic noise but with naturally-occurring errors. Rather than exploring a novel methodology for dealing with label errors, which has been extensively studied in the literature [4], this paper aims to characterize the prevalence of label errors in the test data of popular benchmarks used to measure ML progress and subsequently analyze practical consequences of these errors, and in particular, their effects on model selection. Using *confident learning* [33], we algorithmically identify putative label errors in test sets at scale, and we validate these label errors through human evaluation, estimating a lower-bound of 3.3% errors on average across the 10 datasets. We identify, for example, 2916 (6%) errors in the ImageNet validation set (which is *commonly used as a test set*), and estimate over 5 million (10%) errors in QuickDraw. Figure 1 shows examples of validated label errors for the image datasets in our study.

We use ImageNet and CIFAR-10 as case studies to understand the consequences of test set label errors on benchmark stability. While there are numerous erroneous labels in these benchmarks' test data, we find that relative rankings of models in benchmarks are unaffected after removing or correcting these label errors. However, we find that these benchmark results are *unstable*: higher-capacity models (like NASNet) undesirably reflect the distribution of systematic label errors in their predictions to a greater degree than models with fewer parameters (like ResNet-18), and this effect *increases* with the prevalence of mislabeled test data. This is not traditional overfitting. Larger models are able to

generalize better to the given noisy labels in the test data, but this is problematic because these models produce *worse* predictions than their lower-capacity counterparts when evaluated on the corrected labels for originally-mislabeled test examples.

In real-world settings with high proportions of erroneously labeled data, lower capacity models may thus be practically more useful than their higher capacity counterparts. For example, it may appear NASNet is superior to ResNet-18 based on the test accuracy over originally given labels, but NASNet is in fact worse than ResNet-18 based on the test accuracy over corrected labels. Since the latter form of accuracy is what matters in practice, ResNet-18 should actually be deployed instead of NASNet here — but this is unknowable without correcting the test data labels.

To evaluate how benchmarks of popular pre-trained models change, we incrementally increase the noise prevalence by controlling for the proportion of correctable (but originally mislabeled) data within the test dataset. This procedure allows us to determine, for a particular dataset, at what noise prevalence benchmark rankings change. For example, on ImageNet with corrected labels: ResNet-18 outperforms ResNet-50 if the prevalence of originally mislabeled test examples increases by just 6%.

In summary, our contributions include:

1. The discovery of pervasive label errors in test sets of 10 standard ML benchmarks
2. Open-sourced resources to clean and correct each test set, in which a large fraction of the label errors have been corrected by humans
3. An analysis of the implications of test set label errors on benchmarks, and the finding that higher-capacity models perform better on the subset of incorrectly-labeled test data in terms of their accuracy on the original labels (i.e., what one traditionally measures), but perform worse on this subset in terms of their accuracy on corrected labels (i.e., what one cares about in practice, but cannot measure without the corrected test data we provide)
4. The discovery that merely slight increases in the test label error prevalence would cause model selection to favor the wrong model based on standard test accuracy benchmarks

Our findings imply ML practitioners might benefit from correcting test set labels to benchmark how their models will perform in real-world deployment, and by using simpler/smaller models in applications where labels for their datasets tend to be noisier than the labels in gold-standard benchmark datasets. One way to ascertain whether a dataset is noisy enough to suffer from this effect is to correct at least the test set labels, e.g. using our straightforward approach.

## 2   Background and related work

Experiments in learning with noisy labels [19, 31, 34, 42, 45] suffer a double-edged sword: either synthetic noise must be added to clean training data to measure performance on a clean test set (at the expense of modeling *actual* real-world label noise [18]), or a naturally noisy dataset is used and accuracy is measured on a noisy test set. In the noisy WebVision dataset [24], accuracy on the ImageNet validation data is often reported as a "clean" test set, but several studies [16, 33, 37, 44] have shown the existence of label issues in ImageNet. Unlike these works, we focus exclusively on existence and implications of label errors in the test set, and we extend our analysis to many types of datasets. Although extensive prior work deals with label errors in the *training* set [4, 7], much less work has been done to understand the implications of label errors in the *test set*.

Crowd-sourced curation of labels via multiple human workers [5, 36, 49] is a common method for validating/correcting label issues in datasets, but it can be exorbitantly expensive for large datasets. To circumvent this issue, we only validate subsets of datasets by first estimating which examples are most likely to be mislabeled. To achieve this, we lean on a number of contributions in uncertainty quantification for finding label errors based on prediction/label agreement via confusion matrices [3, 15, 25, 48]; however, these approaches lack either robustness to class imbalance or theoretical support for realistic settings with *asymmetric, non-uniform noise* (for instance, an image of a dog might be more likely to be mislabeled a coyote than a car). For robustness to class imbalance and theoretical support for exact uncertainty quantification, we use a model-agnostic framework, confident learning (CL) [33], to estimate which labels are erroneous across diverse datasets. We choose the CL framework for finding putative label errors because it was empirically found to outperform several recent alternative label error identification methods [23, 33, 46]. Unlike prior work that only

models symmetric label noise [45], we quantify class-conditional label noise with CL, validating the correctable nature of the label errors via crowdsourced workers. Human validation confirms the noise in common benchmark datasets is indeed primarily systematic mislabeling, not just random noise or lack of signal (e.g. images with fingers blocking the camera).

# 3 Identifying label errors in benchmark datasets

Here we summarize our algorithmic label error identification performed prior to crowd-sourced human verification. An overview of each dataset and any modifications is detailed in Appendix A. Step-by-step instructions to obtain each dataset and reproduce the label errors for each dataset are provided at `https://github.com/cleanlab/label-errors`. Our code relies on the implementation of confident learning open-sourced at `https://github.com/cleanlab/cleanlab`. The primary contribution of this section is not in the methodology, which is covered extensively in Northcutt et al. [33], but in its utilization as a *filtering* process to significantly (often as much as 90%) reduce the number of examples requiring human validation in the next step.

To identify label errors in a test dataset with $n$ examples and $m$ classes, we first characterize label noise in the dataset using the confident learning (CL) framework [33] to estimate $\boldsymbol{Q}_{\tilde{y},y^*}$, the $m \times m$ discrete joint distribution of observed, noisy labels, $\tilde{y}$, and unknown, true labels, $y^*$. Inherent in $\boldsymbol{Q}_{\tilde{y},y^*}$ is the assumption that noise is class-conditional [1], depending only on the latent true class, not the data. This assumption is commonly used [9, 32, 42] because it is reasonable. For example, in ImageNet, a *tiger* is more likely to be mislabeled *cheetah* than *CD player*.

The diagonal entry $\hat{p}(\tilde{y}{=}i, y^*{=}i)$ of matrix $\boldsymbol{Q}_{\tilde{y},y^*}$ is the probability that examples in class $i$ are correctly labeled. If the dataset is error-free, then $\sum_{i \in [m]} \hat{p}(\tilde{y}{=}i, y^*{=}i) = 1$. The fraction of label errors is $\rho = 1 - \sum_{i \in [m]} \hat{p}(\tilde{y}{=}i, y^*{=}i)$ and the number of label errors is $\rho \cdot n$. To find label errors, we choose the top $\rho \cdot n$ examples ordered by the normalized margin: $\hat{p}(\tilde{y}{=}i; \boldsymbol{x}) - \max_{j \neq i} \hat{p}(\tilde{y}{=}j; \boldsymbol{x})$ [47]. Table 1 shows the number of CL-guessed label errors for each test set in our study. CL estimation of $\boldsymbol{Q}_{\tilde{y},y^*}$ is summarized in Appendix C.

**Computing out-of-sample predicted probabilities**   Estimating $\boldsymbol{Q}_{\tilde{y},y^*}$ for CL noise characterization requires two inputs for each dataset: (1) out-of-sample predicted probabilities $\hat{\boldsymbol{P}}_{k,i}$ ($n \times m$ matrix) and (2) the test set labels $\tilde{y}_k$. We observe the best results computing $\hat{\boldsymbol{P}}_{k,i}$ by pre-training on the train set, then fine-tuning (all layers) on the test set using cross-validation to ensure $\hat{\boldsymbol{P}}_{k,i}$ is out-of-sample. If pre-trained models are open-sourced (e.g. ImageNet), we use them instead of pre-training ourselves. If the dataset did not have an explicit test set (e.g. QuickDraw and Amazon Reviews), we skip pre-training and compute $\hat{\boldsymbol{P}}_{k,i}$ using cross-validation on the entire dataset. For all datasets, we try common models that achieve reasonable accuracy with minimal hyper-parameter tuning and use the model yielding the highest cross-validation accuracy, reported in Table 1.

Using this approach allows us to find label errors without manually checking the entire test set, because CL identifies potential label errors automatically.

Table 1: Test set errors are prominent across common benchmark datasets. We observe that error rates vary across datasets, from 0.15% (MNIST) to 10.12% (QuickDraw); unsurprisingly, simpler datasets, datasets with more carefully designed labeling methodologies, and datasets with more careful human curation generally had less error than datasets that used more automated data collection procedures.

| Dataset | Modality | Size | Model | Test Set Errors | | | | |
| --- | --- | --- | --- | --- | --- | --- | --- | --- |
| | | | | CL guessed | MTurk checked | validated | estimated | % error |
| MNIST | image | 10,000 | 2-conv CNN | 100 | 100 (100%) | 15 | - | 0.15 |
| CIFAR-10 | image | 10,000 | VGG | 275 | 275 (100%) | 54 | - | 0.54 |
| CIFAR-100 | image | 10,000 | VGG | 2,235 | 2,235 (100%) | 585 | - | 5.85 |
| Caltech-256† | image | 29,780 | Wide ResNet-50-2 | 2,360 | 2,360 (100%) | 458 | | 1.54 |
| ImageNet* | image | 50,000 | ResNet-50 | 5,440 | 5,440 (100%) | 2,916 | - | 5.83 |
| QuickDraw† | image | 50,426,266 | VGG | 6,825,383 | 2,500 (0.04%) | 1870 | 5,105,386 | 10.12 |
| 20news | text | 7,532 | TFIDF + SGD | 93 | 93 (100%) | 82 | - | 1.09 |
| IMDB | text | 25,000 | FastText | 1,310 | 1,310 (100%) | 725 | - | 2.90 |
| Amazon Reviews† | text | 9,996,437 | FastText | 533,249 | 1,000 (0.2%) | 732 | 390,338 | 3.90 |
| AudioSet | audio | 20,371 | VGG | 307 | 307 (100%) | 275 | - | 1.35 |

*Because the ImageNet test set labels are not publicly available, the ILSVRC 2012 validation set is used.
†Because no explicit test set is provided, we study the entire dataset to ensure coverage of any train/test split.

Table 2: Mechanical Turk validation of CL-flagged errors and categorization of label issues.

| Dataset | Test Set Errors Categorization | | | | | |
| --- | --- | --- | --- | --- | --- | --- |
| | non-errors | errors | non-agreement | correctable | multi-label | neither |
| MNIST | 85 | 15 | 2 | 10 | - | 3 |
| CIFAR-10 | 221 | 54 | 32 | 18 | 0 | 4 |
| CIFAR-100 | 1650 | 585 | 210 | 318 | 20 | 37 |
| Caltech-256 | 1902 | 458 | 99 | 221 | 115 | 23 |
| ImageNet | 2524 | 2916 | 598 | 1428 | 597 | 293 |
| QuickDraw | 630 | 1870 | 563 | 1047 | 20 | 240 |
| 20news | 11 | 82 | 43 | 22 | 12 | 5 |
| IMDB | 585 | 725 | 552 | 173 | - | - |
| Amazon Reviews | 268 | 732 | 430 | 302 | - | - |
| AudioSet | 32 | 275 | - | - | - | - |

## 4 Validating label errors with Mechanical Turk

We validated the algorithmically identified label errors with a Mechanical Turk (MTurk) study. For two large datasets with a large number of errors (QuickDraw and Amazon Reviews), we checked a random sample; for the rest, we checked all identified errors.

We presented workers with hypothesized errors and asked them whether they saw the (1) given label, (2) the top CL-predicted label, (3) both labels, or (4) neither label in the example. To aid the worker, the interface included high-confidence examples of the given class and the CL-predicted class. Figure S1 in Appendix B shows a screenshot of the MTurk worker interface.

Each CL-flagged label error was independently presented to five workers. We consider the example validated (an "error") if fewer than three of the workers agree that the data point has the given label (*agreement threshold = 3 of 5*), otherwise we consider it to be a "non-error" (i.e. the original label was correct). We further categorize the label errors, breaking them down into (1) "correctable", where a majority agree on the CL-predicted label; (2) "multi-label", where a majority agree on both labels appearing; (3) "neither", where a majority agree on neither label appearing; and (4) "non-agreement", a catch-all category for when there is no majority. Table 2 summarizes the results, and Figure 1 shows examples of validated label errors from image datasets.

### 4.1 Failure modes of confident learning

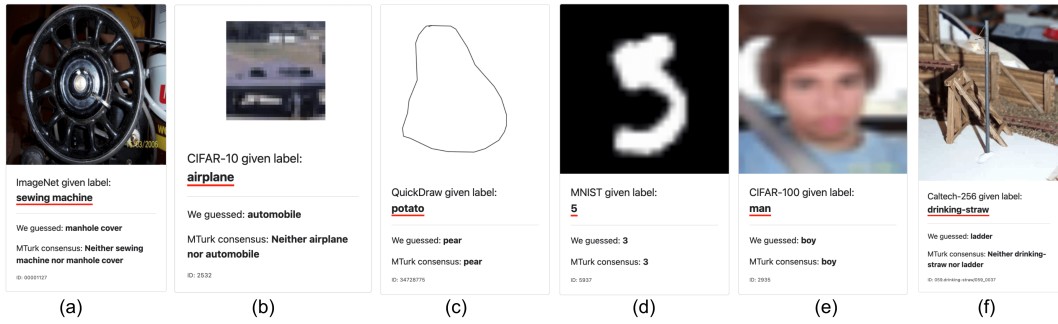

Figure 2: Difficult examples from various datasets where confident learning finds a potential label error but human validation shows that there actually is no error. Example (a) is a cropped image of part of an antiquated sewing machine; (b) is a viewpoint from inside an airplane, looking out at the runway and grass with a partial view of the nose of the plane; (c) is an ambiguous shape which could be a potato; (d) is likely a badly drawn "5"; (e) is a male whose exact age cannot be determined; and (f) is a straw used as a pole within a miniature replica of a village.

Confident learning sometimes flags data points that are not actually erroneous. By visually inspecting putative label errors, we identified certain previously unexamined failure modes of confident learning [33]. Appendix D provides a mathematical description of the conditions under which these failure modes occur. Figure 2 shows uniquely challenging examples, with excessively erroneous $\hat{p}(\tilde{y}=j; \boldsymbol{x})$, where failure mode cases potentially occur. The sewing machine in Figure 2(a), for example, exhibits a "part versus whole" issue where the image has been cropped to only show a small portion of the object. The airplane in Figure 2(b) is an unusual example of the class, showing the plane from the perspective of the pilot looking out of the front cockpit window.

Figure 2 clarifies that our corrected test set labels are not 100% perfect. Even with a stringent 5 of 5 agreement threshold where all human reviewers agreed on a label correction, the "corrected" label is not always actually correct. Fortunately, these failure mode cases are rare. Inspection of https://labelerrors.com shows that the majority of the labels we corrected are reasonable. Our corrected test sets, while imperfect in these cases, are improved from the original test sets.

## 5 Implications of label errors in test data

Finally, we consider how pervasive test set label errors may affect ML practitioners in real-world applications. To clarify the discussion, we first introduce some useful terminology.

**Definition 1** (original accuracy, $\tilde{A}$). *The accuracy of a model's predicted labels over a given dataset, computed with respect to the original labels present in the dataset. Measuring $\tilde{A}$ over the test set is the standard way practitioners evaluate their models today.*

**Definition 2** (corrected accuracy, $A^*$). *The accuracy of a model's predicted labels, computed over a modified dataset in which previously identified erroneous labels have been corrected by humans to the true class when possible and removed when not. Measuring $A^*$ over the test set is preferable to $\tilde{A}$ for evaluating models because $A^*$ better reflects performance in real-world applications.*

The *human* labelers referenced throughout this section are the workers from our MTurk study in Section 4. In the following definitions, \ denotes a set difference and $\mathcal{D}$ denotes the full test dataset.

**Definition 3** (benign set, $\mathcal{B}$). *The subset of benign test examples, comprising data that CL did not flag as likely label errors and data that was flagged but for which human reviewers agreed that the original label should be kept. ($\mathcal{B} \subset \mathcal{D}$)*

**Definition 4** (unknown-label set, $\mathcal{U}$). *The subset of CL-flagged test examples for which human labelers could not agree on a single correct label. This includes examples where human reviewers agreed that multiple classes or none of the classes are appropriate. ($\mathcal{U} \subset \mathcal{D} \backslash \mathcal{B}$)*

**Definition 5** (pruned set, $\mathcal{P}$). *The remaining test data after removing $\mathcal{U}$ from $\mathcal{D}$. ($\mathcal{P} = \mathcal{D} \backslash \mathcal{U}$)*

**Definition 6** (correctable set, $\mathcal{C}$). *The subset of CL-flagged examples for which human-validation reached consensus on a different label than the originally given label. ($\mathcal{C} = \mathcal{P} \backslash \mathcal{B}$)*

**Definition 7** (noise prevalence, $N$). *The percentage of the pruned set comprised of the correctable set, i.e. what fraction of data received the wrong label in the original benchmark when a clear alternative ground-truth label should have been assigned (disregarding any data for which humans failed to find a clear alternative). Here we operationalize noise prevalence as $N = \frac{|\mathcal{C}|}{|\mathcal{P}|}$.*

These definitions imply $\mathcal{B}, \mathcal{C}, \mathcal{U}$ are disjoint with $\mathcal{D} = \mathcal{B} \cup \mathcal{C} \cup \mathcal{U}$ and also $\mathcal{P} = \mathcal{B} \cup \mathcal{C}$. In subsequent experiments, we report corrected test accuracy over $\mathcal{P}$ after correcting all of the labels in $\mathcal{C} \subset \mathcal{P}$. We ignore the unknown-label set $\mathcal{U}$ (and do not include those examples in our estimate of noise prevalence) because it is unclear how to measure *corrected accuracy* for examples whose true underlying label remains ambiguous. Thus the *noise prevalence* reported throughout this section differs from the fraction of label errors originally found in each of the test sets.

A major issue in ML today is that one only sees the original test accuracy in practice, whereas one would prefer to base modeling decisions on the corrected test accuracy instead. Our subsequent discussion highlights the potential implications of this mismatch. What are the consequences of test set label errors? Figure 3 compares performance on the ImageNet validation set, *commonly used in place of the test set*, of 34 pre-trained models from the PyTorch and Keras repositories (throughout, we use provided checkpoints of models that have been fit to the original training set). Figure 3a confirms the observations of Recht et al. [37]; benchmark conclusions are largely unchanged by using a corrected test set, i.e. in our case by removing errors.

### 5.1 Benchmarking on the correctable set

However, we find a surprising result upon closer examination of the models' performance on the correctable set $\mathcal{C}$. When evaluating models *only* on these originally-mislabeled test data, models which perform best on the original (incorrect) labels perform the worst on the corrected labels. For example, ResNet-18 [14] significantly outperforms NASNet [50] in terms of corrected accuracy

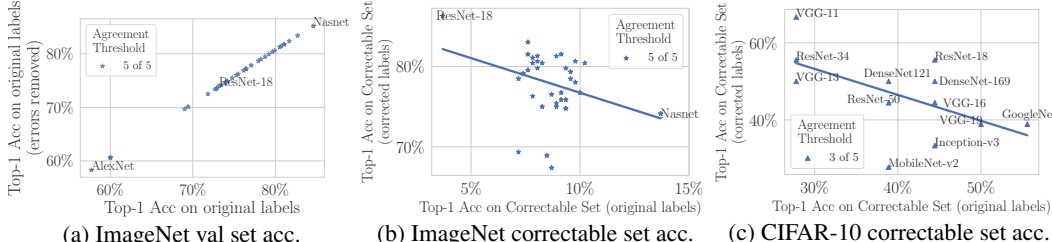

(a) ImageNet val set acc.     (b) ImageNet correctable set acc.     (c) CIFAR-10 correctable set acc.

Figure 3: Benchmark ranking comparison of 34 pre-trained ImageNet models and 13 pre-trained CIFAR-10 models (details in Tables S2 and S1 and Figure S3 in the Appendix). Benchmarks are unchanged by removing label errors (a), but change drastically (b) on the Correctable set with original (erroneous) labels versus corrected labels, e.g. NASNet: 1/34 → 29/34, ResNet-18: 34/34 → 1/34.

over $\mathcal{C}$, despite exhibiting far worse original test accuracy. The change in ranking can be dramatic: NASNet-large drops from ranking 1/34 → 29/34, Xception drops from ranking 2/34 → 24/34, ResNet-18 increases from ranking 34/34 → 1/34, and ResNet-50 increases from ranking 20/24 → 2/24 (see Table S1 in the Appendix). We verified that the same trend occurs independently across 13 pre-trained CIFAR-10 models (Figure 3c), e.g. VGG-11 significantly outperforms VGG-19 [41] in terms of corrected accuracy over $\mathcal{C}$. Note that all numbers reported here are over subsets of the held-out test data, so this is not overfitting in the classical sense.

This phenomenon, depicted in Figures 3b and 3c, may indicate two key insights: (1) lower-capacity models provide unexpected regularization benefits and are more resistant to learning the asymmetric distribution of noisy labels, (2) over time, the more recent (larger) models have architecture/hyperparameter decisions that were made on the basis of the (original) test accuracy. Learning to capture systematic patterns of label error in their predictions allows these models to improve their original test accuracy, but this is not the sort of progress ML research should aim to achieve. Harutyunyan et al. [13] and Arpit et al. [2] have previously analyzed phenomena similar to (1), and here we demonstrate that this issue really does occur for the models/datasets widely used in current practice. (2) is an undesirable form of overfitting, albeit not in the classical sense (as the original test accuracy can further improve through better modeling of label errors), but rather overfitting to the specific benchmark (and quirks of the original label annotators) such that accuracy improvements for erroneous labels may not translate to superior performance in a deployed ML system.

This phenomenon has important practical implications for real-world datasets with greater noise prevalence than the highly curated benchmark data studied here. In these relatively clean benchmark datasets, the noise prevalence is an underestimate as we could only verify a subset of our candidate label errors rather than all test labels, and thus the potential gap between original vs. corrected test accuracy over $\mathcal{P}$ is limited for these particular benchmarks. However, this gap increases proportionally for data with more (correctable) label errors in the test set, i.e. as $N$ increases.

## 5.2 Benchmark instability

To investigate how benchmarks of popular models change with varying proportions of label errors in test sets, we randomly and incrementally remove correctly-labeled examples, one at a time, until only the original set of mislabeled test data (with corrected labels) is left. We create alternate versions (subsets) of the pruned benchmark test data $\mathcal{P}$, in which we additionally randomly omit some fraction, $x$, of $\mathcal{B}$ (the non-CL-flagged test examples). This effectively increases the proportion of the resulting test dataset comprised of the correctable set $\mathcal{C}$, and reflects how test sets function in applications with greater prevalence of label errors. If we remove a fraction $x$ of benign test examples (in $\mathcal{B}$) from $\mathcal{P}$, we estimate the noise prevalence in the new (reduced) test dataset to be $N = \frac{|\mathcal{C}|}{|\mathcal{P}| - x|\mathcal{B}|}$. By varying $x$ from 0 to 1, we can simulate any noise prevalence ranging from $|\mathcal{C}|/|\mathcal{P}|$ to 1. We operationalize averaging over all choices of removal by linearly interpolating from accuracies over the corrected test set ($\mathcal{P}$, with corrected labels for the subset $\mathcal{C}$) to accuracies over the erroneously labeled subset ($\mathcal{C}$, with corrected labels). Over these corrected test sets, we evaluate popular pre-trained models (again using provided checkpoints of models that have been fit to the original training set).

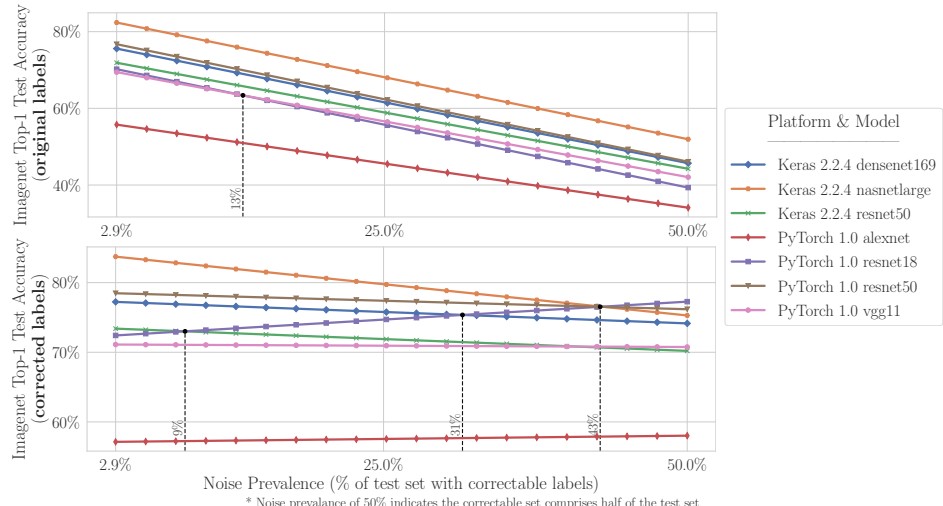

Figure 4: ImageNet top-1 original accuracy (top) and corrected accuracy (bottom) vs noise prevalence (agreement threshold = 3). Vertical lines indicate noise levels at which the ranking of two models changes (in terms of original/corrected accuracy). The left-most point ($N = 2.9\%$) on the x-axis is $|\mathcal{C}|/|\mathcal{P}|$, i.e. the (rounded) estimated noise prevalence of the pruned set, $\mathcal{P}$. The leftmost vertical dotted line in the bottom panel is read, "The ResNet-50 and ResNet-18 benchmarks cross at noise prevalence $N = 9\%$," implying ResNet-18 outperforms ResNet-50 when $N$ increases by around $6\%$ relative to the original pruned test data ($N = 2.9\%$ originally, c.f. Table 2).

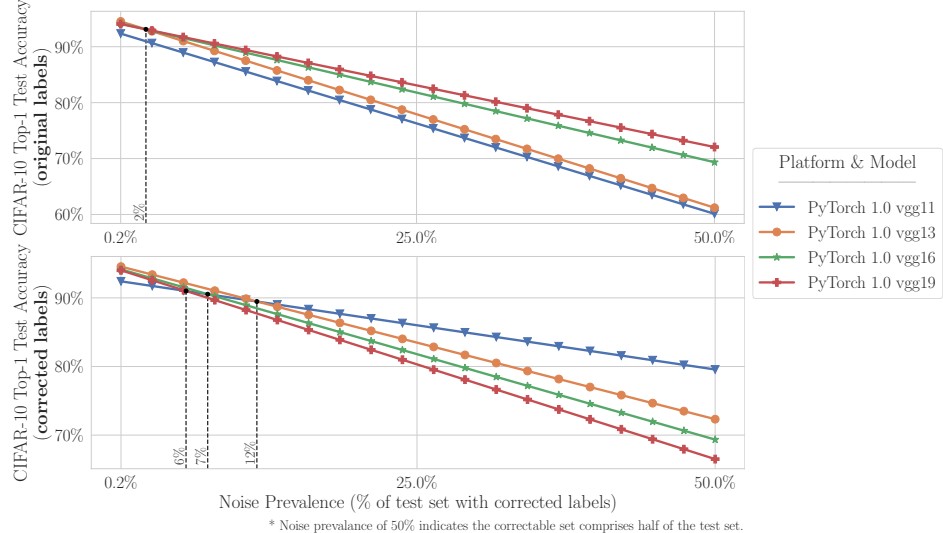

Figure 5: CIFAR-10 top-1 original accuracy (top panel) and corrected accuracy (bottom panel) vs Noise Prevalence (agreement threshold = 3). For additional details, see the caption of Figure 4.

For a given model $\mathcal{M}$, its resulting accuracy (as a function of $x$) over the reduced test data is given by $A(x; \mathcal{M}) = \frac{A_{\mathcal{C}}(\mathcal{M}) \cdot |\mathcal{C}| + (1-x) \cdot A_{\mathcal{B}}(\mathcal{M}) \cdot |\mathcal{B}|}{|\mathcal{C}| + (1-x) \cdot |\mathcal{B}|}$, where $A_{\mathcal{C}}(\mathcal{M})$ and $A_{\mathcal{B}}(\mathcal{M})$ denote the (original or corrected) accuracy over the correctable set and benign set, respectively (accuracy before removing any examples). Here $A_{\mathcal{B}} = A_{\mathcal{B}}^* = \tilde{A}_{\mathcal{B}}$ because no erroneous labels were identified in $\mathcal{B}$. The expectation is taken over which fraction $x$ of examples are randomly removed from $\mathcal{B}$ to produce the reduced test set: the resulting expected accuracy, $A(x; \mathcal{M})$, is depicted on the y-axis of Figures 4-5. As our removal of test examples was random from the non-mislabeled set, we expect this reduced test data is representative of test sets that would be used in applications with a similarly greater prevalence

of label errors. Note that we ignore non-correctable data with unknown labels ($\mathcal{U}$) throughout this analysis, as it is unclear how to report a better version of the accuracy for such ill-specified examples.

Over alternative (reduced) test sets created by imposing increasing degrees of noise prevalence in ImageNet/CIFAR-10, Figures 4-5 depict the resulting original (erroneous) test set accuracy and corrected accuracy of the models, expected on each alternative test set. For a given test set (i.e. point along the $x$-axis of these plots), the vertical ordering of the lines indicates how models would be favored based on original accuracy or corrected accuracy over this test set. Unsurprisingly, we see that more flexible/recent architectures tend to be favored on the basis of original accuracy, regardless of which test set (of varying noise prevalence) is considered. This aligns with conventional expectations that powerful models like NASNet will outperform simpler models like ResNet-18. However, if we shift our focus to the corrected accuracy (i.e. what actually matters in practice), it is no longer the case that more powerful models are reliably better than their simpler counterparts: the performance strongly depends on the degree of noise prevalence in the test data. For datasets where label errors are common, a practitioner is more likely to select a model (based on original accuracy) that is not actually the best model (in terms of corrected accuracy) to deploy.

Finally, we note that this analysis only presents a loose lower bound on the magnitude of these issues due to unaccounted for label errors in the non-CL-flagged data (see Section 6). We only identified a subset of the actual correctable set as we are limited to human-verifiable label corrections for a subset of data candidates (algorithmically prioritized via confident learning). Because the actual correctable sets are likely larger, our noise prevalence estimates are optimistic in favor of higher capacity models. Thus, the true gap between corrected vs. original accuracy may be larger and of greater practical significance, even for the gold-standard benchmark datasets considered here. For many application-specific datasets collected by ML practitioners, the noise prevalence will be greater than the numbers presented here: thus, it is imperative to be cognizant of the distinction between corrected vs. original accuracy, and to utilize careful data curation practices, perhaps by allocating more of an annotation budget to ensure higher quality labels in the test data.

## 6 Expert review of CL-flagged and non-CL-flagged label errors in ImageNet

Up to this point, we have only evaluated the subsets of the datasets flagged by CL: how do we know that CL-flagged examples are indeed more erroneous than a random subset of a dataset? How many label errors are missed in the non-CL-flagged data? And how reliable are MTurk workers in comparison to expert reviewers? In this section, we address these questions by conducting an additional expert review of both CL-flagged and non-CL-flagged examples in the ImageNet val set.

The expert review was conducted in two phases (details in Appendix G). In the first phase, experts reviewed 1 randomly-selected CL-flagged example and 1 randomly-selected non-CL-flagged example from each of the 1,000 ImageNet classes (66 classes had no CL-flagged example, i.e. 934 + 1,000 = 1934 images were evaluated in total). Given a similar interface as MTurk workers, the expert reviewers selected one choice from: (1) the given label, (2) the top-most predicted label that differs from the given label, (3) "both", and (4) "neither". Experts researched any unfamiliar classes by looking up related images and taxonomy information online, spending on average 13x more time per label than MTurk workers. Each image was reviewed by at least two experts, and experts agreed on decisions for 77% of the images. In the second phase, all experts jointly revisited the remaining 23% where there was disagreement and came to a consensus on a single choice.

Table 3 reveals that the set of CL-flagged examples has significantly higher proportions of every type of label issue than the set of non-CL-flagged examples. An image flagged by CL was 2.6x as likely to be erroneously labeled than an non-CL-flagged image. Given a limited budget for human review, we thus recommend using CL to prioritize examples when verifying the labels in a large dataset.

Comparing *CL (expert)* to *CL (MTurk)* in Table 3 indicates that for CL-flagged examples, MTurk workers favored correcting labels in cases where experts agreed neither label was appropriate. For this analysis, we only consider the subset of MTurk reviewed images that overlaps with the 1,934 expert reviewed images. This may be attributed to experts knowing a better choice than the two label choices presented in the task (c.f. Figure S2). Nonetheless the MTurk results overall agree with those from our expert review. This validates our overall approach of using CL followed by MTurk to characterize label errors, and demonstrates that a well-designed interface (Figure S1) suffices for non-expert workers to provide high-quality label verification of datasets.

Table 3: Percentages of label errors identified by experts vs. MTurk workers in CL-flagged examples and random non-CL-flagged examples from ImageNet. Only experts reviewed non-CL examples. The first two rows are computed over the same subset of images. The last column lists average time spent reviewing each image. Percentages are row-normalized, with raw counts provided in Table S3.

|               | non-errors | errors | correctable | multi-label | neither | Avg. time spent |
|---------------|------------|--------|-------------|-------------|---------|-----------------|
| CL (MTurk)    | 57.9%      | 42.2%  | 24.7%       | 11.1%       | 6.4%    | 5 seconds       |
| CL (expert)   | 58.7%      | 41.4%  | 17.7%       | 13.1%       | 10.6%   | 67 seconds      |
| non-CL (expert) | 84.0%    | 16.0%  | 3.2%        | 9.1%        | 3.7%    | 67 seconds      |

We further estimate that the analysis in previous sections missed around 14% of the label errors in ImageNet because 89% of images were not flagged by CL and Table 3 indicates around 16% of these were mislabeled. By including the additional 14% error found from the *9x larger* set of non-CL-flagged examples, we can more accurately estimate that the ImageNet validation set contains closer to 20% label errors (up from the 6% reported in Table 1). This roughly indicates *how much more* severe the issue of label errors actually is compared to what we reported in Sections 4 and 5.

## 7   Discussion

This paper demonstrates that label errors are ubiquitous in the test sets of many popular benchmarks used to gauge progress in machine learning. We hypothesize that this has not been previously discovered and publicized at such scale due to various challenges. Firstly, human verification of all labels can be quite costly, which we circumvented here by using CL algorithms to filter automatically for likely label errors prior to human verification. Secondly, working with 10 differently formatted datasets was nontrivial, with some exhibiting peculiar issues upon close inspection (despite being standard benchmarks). For example, IMDB, QuickDraw, and Caltech-256 lack a global index making it difficult to map model outputs to corrected test examples on different systems. We provide index files in our repository[1] to address this. Furthermore, Caltech-256 contains several duplicate images, of which which we found no previous mention. Lastly, ImageNet contains duplicate class labels, e.g. "maillot" (638 & 639) and "crane" (134 & 517) [33, 44].

Traditionally, ML practitioners choose which model to deploy based on test accuracy — our findings advise caution here. Instead, judging models over correctly labeled test sets may be important, especially for real-world datasets that are likely noisier than these popular benchmarks. Small increases in the prevalence of mislabeled test data can destabilize ML benchmarks, indicating that low-capacity models may actually outperform high-capacity models in noisy real-world applications, even if their measured performance on the original test data appears worse. We recommend considering the distinction between corrected vs. original test accuracy and curating datasets to maximize high-quality test labels, even if budget constraints only allow for lower-quality training labels.

This paper shares new findings about pervasive label errors in test sets and their effects on benchmark stability, but it does not address whether the apparent overfitting of high-capacity models versus low-capacity models is due to overfitting to train set noise, overfitting to validation set noise during hyper-parameter tuning, or heightened sensitivity to train/test label distribution shift that occurs when test labels are corrected. An intuitive hypothesis is that high-capacity models more closely fit all statistical patterns present in the data, including those patterns related to systematic label errors that models with more limited capacity are less capable of closely approximating. A rigorous analysis to disambiguate and understand the contribution of each of these causes and their effects on benchmarking stability is a natural next step, which we leave for future work. How to best allocate a given human label verification budget between training and test data also remains an open question.

## Acknowledgments

This work was supported in part by funding from the MIT-IBM Watson AI Lab. We thank Jessy Lin for her contributions to early stages of this research, and we thank Wei Jing Lok for his contributions to the ImageNet expert labeling experiments.

---

[1]https://github.com/cleanlab/label-errors#how-to-download-prepare-and-index-the-datasets

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
