# OpenReview forum: "Pervasive Label Errors in Test Sets Destabilize Machine Learning Benchmarks"
_NeurIPS.cc/2021/Track/Datasets_and_Benchmarks/Round1 — NeurIPS 2021 Datasets and Benchmarks Track (Round 1)_

### Official Review · Reviewer_bcPL · 2021-06-30
**Label Errors in Test Sets**

**Rating:** 8
**Confidence:** 4
**Clarity:** The paper is very well written, no re…

**Strengths:**

The problem of label errors in test sets has not been given enough attention so far and the authors are filling an important gap here: they use confident learning to find errors in test sets. In paper, it is shown that such errors can have important implications when models are compared.

**Weaknesses:**

The main weakness of the paper is that the "dark matter" of hard-to-detect label errors is not considered and discussed. It might be the case that such errors have different impact for simpler and complicated models. I think the authors should run their procedure for the manual identification of label errors on random samples as an additional step to compare the statistics with the errors found with the help of confident learning.

Minor issue: The label "flying saucer" in an example given in Figure 1 is quite arbitrary, I'd say. I think it merit some discussion, problems like this.

Another issue is that the variance of error rate across the 10 test sets is quite large (10% for Quick Draw vs <2% for 3 data sets), this should be discussed.

**Additional Feedback:**

Minor remark:

* 178: spelling "practioners"

**Correctness:**

In general, the submission is correct, though see the discussion of confounders in Weaknesses.

**Documentation:**

The data are well documented and organized.

**Ethics:**

No ethical concerns.

**Relation To Prior Work:**

I did not find any issues here.

**Summary And Contributions:**

This is an important and interesting work concerning label errors in commonly used benchmarks.

Contributions

* analysis of label errors in test sets of 10 commonly-used datasets
* release of cleaned tests for the data sets
* analysis of implications of label errors in benchmarking neural models

---

> ### Comment · Reviewer_bcPL · 2021-07-14
> **Note**
>
> Note that the main weakness given by me is the same as Issue 2 given by reviewer v4vb. I still think this is the biggest problem in the paper.

---

> > ### Author Response · Authors · 2021-07-14
> > **The main points of feedback have been addressed with the addition of an expert validation experiment**
> >
> > Thank you for your update and for your review. We agree with your (and reviewer v4vb's) assessment and are finishing up a random (non-CL-identified) small scale expert validation experiment. Adding an additional expert validation experiment (conducted carefully by a small number of human expert raters) is time-consuming hence our following up on the final day of the rebuttal.
> >
> > We further conducted the same expert validation experiment on a subset of CL-identified examples to validate that our expert experiment findings agree with the mechanical turk experiment findings.
> >
> > We believe these two additions to the work address the two major points of feedback brought in this review process.
> >
> > Again, a big thanks to the reviewers for bringing these points to our attention. We look forward to sharing the results later today, along with individual responses for each reviewer.
> >
> > **EDIT: Our primary findings are now available in the response to Reviewer v4vb.**

---

> ### Author Response · Authors · 2021-07-15
> **Thank you for the suggestions and the positive review. We have responded about the “dark matter” in the response to Reviewer v4vb. We address the remaining feedback below.**
>
> Thank you for the suggestions and the positive review. To avoid duplication, we have responded to the question about the “dark matter” of label errors in the response to Reviewer v4vb. We address the remaining feedback below.
>
> > Minor issue: The label "flying saucer" in an example given in Figure 1 is quite arbitrary, I'd say. I think it merits some discussion, problems like this.
>
> Thank you for pointing this out! We think that this is quite interesting indeed, and it’s a class of label error that’s not currently discussed in detail in the paper as it stands: a multi-label data point that doesn’t “contain both classes”, but “can be interpreted as either class”. Another interesting example of a “type” of error we found in a different dataset, ImageNet: an image is labeled as “corn”, but the image is of a pot of vegetable soup; the soup has a kernel of corn visible, but the corn takes up about 1% of the image area. Should the image be labeled as “corn”, “soup”, or both? Through looking at the data in these datasets informally over the course of our work, as well as through the “expert validation” experiment described in the response to Reviewer v4vb, we have seen many examples of many types of errors. We will add discussion on how there are many failure modes, and give some examples of some of them, but a full characterization seems beyond the scope of this paper. We think it would certainly be interesting future work.
>
> > Another issue is that the variance of error rate across the 10 test sets is quite large (10% for Quick Draw vs <2% for 3 data sets), this should be discussed.
>
> Higher error rates appear in “more complex” datasets, with more classes/subtlety, and ones that are “less carefully curated”, with more automation and less human/expert input. For example, QuickDraw (high error rate) asked humans to draw certain things, but there was no further review of images drawn by individual humans. On the other hand, ImageNet (lower error rate) collected images automatically but then had multiple Mechanical Turk workers contribute to labeling each image. We have summarized the dataset curation process for each dataset we study in Appendix A. We will make an explicit connection between error rate and curation methodology in the body of the paper and add a pointer to the appendix.

---

### Official Review · Reviewer_dsSp · 2021-07-01
**An excellent submission that may compete for the best paper of this track**

**Rating:** 10
**Confidence:** 4
**Correctness:** yes
**Clarity:** yes

**Strengths:**

1) This paper, for the first time, studied the pervasive existences and important consequences of test-data errors in 10 very popular datasets, including ImageNet, CIFAR-10, MNIST, etc.
2) The paper finally produced data-error corrections in these ten important datasets, which could benefit the ML/CV/NLP communities very much.
3) This paper uses the error-data corrections to reveal some important consequences, including a) big-capacity networks may fit too much to the training data-errors; b) some simpler network, e.g. ResNet-18, may work better than ResNet-50, over the corrected error-data. All of these conclusions are super important to the field.
4) This paper is very well-written, easy-to-read, ready-for-publication.
5) The authors also provided the codebase for reproducing the results, a website for data browsing, detailed statistics/data annotation process, etc..

**Weaknesses:**

I don't think this paper has major weaknesses. Below are some small suggestions:
1) For QuickDraw and AmazonReviews, the authors can discuss how do you plan to finish them? or, any other ways to scale up the verification?
2) The information in the current Sec. 5 is too dense. To make it easier to read, I suggest the authors break the materials into some sub-sections or paragraphs.
3) Is it planned to also fix the annotation errors in the training split? Is the same CL framework scalable to fix errors in the training set?

**Additional Feedback:**

no

Post-rebuttal: I want to keep my rating.

**Documentation:**

yes

**Relation To Prior Work:**

yes

**Summary And Contributions:**

This paper, for the first time, investigates the pervasive existence and important consequences of test-data errors in 10 popular ML datasets, including ImageNet, MNIST, CIFAR-10, etc. The paper uses confident learning techniques to propose test-data error candidates and then asked mechanical turkers to confirm or verify the data errors. The contribution of test-data error correction is very important to the ML/vision/NLP fields as these datasets are so popularly used. This paper also produces a significantly important observation that some simpler networks (ResNet-18) sometimes work better than complicated ones (ResNet-50) over data with error corrections, which reveals an important fact that the big-capacity models may fit too well to the training split, even to the data errors there. Overall, this is a very strong submission (and very impactful to the field) that I would even consider as a strong candidate competing for the best paper award of this track.

---

> ### Author Response · Authors · 2021-07-15
> **Thank you for the feedback and favorable review. We address your specific questions/concerns below.**
>
> Thank you for the feedback and the favorable review. We are glad you enjoyed the paper. We address your specific questions/concerns below.
>
> > For QuickDraw and AmazonReviews, the authors can discuss how do you plan to finish them? or, any other ways to scale up the verification?
>
> The cost per data point of validation is the same with Quickdraw/Amazon as with the other datasets, so in that sense, the methodology does scale to those larger datasets. We lacked the funding to validate those larger datasets completely. If the goal was to correct as many data points as possible, one way to get more mileage out of a fixed amount of funding is to validate the K “most promising” potential label errors (i.e. data ordered by normalized margin, see Sec 3) rather than validating K points uniformly sampled from the putative label errors ordered by normalized margin. We did the latter because our goal was to validate the CL-based methodology and obtain an estimate of the total number of label errors.
>
> > The information in the current Sec. 5 is too dense. To make it easier to read, I suggest the authors break the materials into some sub-sections or paragraphs.
>
> Agreed, the section is quite long without any sub-headings. We will add sub-headings in the next revision.
>
> > Is it planned to also fix the annotation errors in the training split? Is the same CL framework scalable to fix errors in the training set?
>
> For this work, we decided to focus on test sets and the implications of errors in the test set. Our methodology also works for correcting errors in any dataset, and would suffice to correct errors in the training set as well. We will mention this in the paper. The implications of training set errors are different, and not discussed in this paper. Train set errors are studied extensively in prior work (see Sec. 2), but no prior work has studied label errors in test sets at scale (across many datasets and modalities).

---

### Official Review · Reviewer_v4vb · 2021-07-05
**Solid work with interesting findings and potential high impacts**

**Rating:** 7
**Confidence:** 4

**Strengths:**

1. It is interesting to find and validate the label errors in test sets of popular benchmarks. The open-source corrected test labels would be impactful.
2. Correcting all the test labels by brute force is expensive. Thus implementing pre-selection algorithmically as employed in this paper is important. Besides, it is worthwhile to deal with real-world human-level label noise rather than synthetic label noise.
3. The paper corrects test labels for many popular benchmark datasets that are heavily used in the literature.


**Weaknesses:**

1. Only a subset of the label errors had been corrected by humans. It is unclear how many label errors are missed. A complete set of corrected labels should include all label errors.
2. The author only validates the label errors guessed by confident learning, making the finding on model capacity not convincing enough. This is because the label errors guessed by confident learning may not be independent. It is unclear whether we can get the same finding if we have a subset of corrected labels independently sampled from the whole set with wrong annotations. Thus, the currently used corrected labels in Fig. 3b may not represent the property of the other undetected label errors. An experiment to show the difference between the real label errors and the current subset may be necessary.


**Additional Feedback:**

1. The paper would be better presented if the above weakness could be fixed.
2. It would be helpful if the author can provide more intuitions on why a larger model performs worse in the corrected label set.
3. Other label error detection methods may also be helpful.


**Clarity:**

Yes. The paper is well-organized and easy to follow.


**Correctness:**

Most of the claims are correct. But it is unclear whether the selected subset of label errors can represent the other undetected label errors. Please check the details in weakness.

**Documentation:**

The datasets are well-documented in the URLs provided in this paper to the best of the reviewer’s knowledge.

**Ethics:**

The authors re-collect test labels for open-source datasets that are primarily standard object recognition tasks. The reviewer hasn’t found ethical concerns yet.

**Relation To Prior Work:**

The paper well positioned in the literature of learning with noisy labels. The authors also did a great job motivating the current study and justified why this is an important missing piece in the literature.

**Summary And Contributions:**

This paper identifies label errors in test sets of popular vision, text, and audio datasets using confident learning. The detected label errors are validated by crowdsourced workers from Amazon MTurk. It is important to ensure all the test labels in benchmark datasets are correct. The core finding in this paper challenges fundamentally the reported benchmark results in the literature so far. The finding that, higher capacity models generalize well on the original incorrect test labels but generalize worse on the corrected test labels than simpler counterparts, is interesting.

---

> ### Author Response · Authors · 2021-07-15
> **We added a small-scale validation experiment to study the proportion of label errors in non-CL-identified examples and address all points of feedback**
>
> Thank you for your feedback. Below, we address your questions/concerns, including the central question about the label errors guessed by confident learning versus label errors in general, a point also raised by Reviewer bcPL.
>
> > The author only validates the label errors guessed by confident learning, making the finding on model capacity not convincing enough. This is because the label errors guessed by confident learning may not be independent. It is unclear whether we can get the same finding if we have a subset of corrected labels independently sampled from the whole set with wrong annotations. Thus, the currently used corrected labels in Fig. 3b may not represent the property of the other undetected label errors. An experiment to show the difference between the real label errors and the current subset may be necessary.
>
> And related feedback from Reviewer bcPL:
>
> > The main weakness of the paper is that the "dark matter" of hard-to-detect label errors is not considered and discussed. It might be the case that such errors have different impact for simpler and complicated models. I think the authors should run their procedure for the manual identification of label errors on random samples as an additional step to compare the statistics with the errors found with the help of CL.
>
> This was great feedback. We conducted a small-scale expert validation experiment to address each of these points and agree that its inclusion strengthened the work -- thanks for bringing this to our attention. In this additional experiment, three ‘expert’ reviewers evaluated both CL-identified and random non-CL-identified examples/images from ImageNet in a similar format as our Mechanical Turk experiment. As with the mTurk experiment, the expert raters select one choice from: (1) the given label, (2) the top-most predicted label which differs from the given label, (3) ‘both’ and (4) ‘neither’. 2100 examples were reviewed (1093 non-CL-identified and 1007 CL-identified). Raters were asked to research any discrepancies for which they were unfamiliar by looking up related images online and reviewing taxonomy materials. On average, raters spent 54 seconds on each example.
>
> The addition of this validation experiment serves three purposes:
> - To study how useful CL is for flagging flawed labels for human review (do we find the same amount of error if we validate examples which were not found via CL?)
> - To provide a baseline for expert review of CL (in case it differs from mechanical Turk and for a fair comparison with the previous point).
> - To validate the quality of errors found via our mechanical Turk validation experiment.
>
> Here, we present our preliminary results completed within the timeframe of this rebuttal cycle:
>
> TABLE 1 - counts of label errors identified by experts in CL-flagged examples vs non-CL flagged random examples, where count(errors) = count(correctable) + count(multi-label) + count(neither)
>
> |                 |   total |   non-errors |   errors |   correctable |   multi-label |   neither |
> |:----------------|--------:|-------------:|---------:|--------------:|--------------:|----------:|
> | CL (expert)     |    1007 |          578 |      429 |           211 |           122 |        96 |
> | non-CL (expert) |    1093 |          892 |      201 |            70 |            76 |        55 |
> | total           |    2100 |         1470 |      630 |           281 |           198 |       151 |
>
> TABLE 2 - percentages of label errors identified by experts vs mTurkers in CL-flagged examples and non-CL flagged random examples (only experts reviewed non-CL examples). Percentages are row-normalized
>
> |                 | non-errors   | errors   | correctable   | multi-label   | neither   |
> |:----------------|:-------------|:---------|:--------------|:--------------|:----------|
> | CL (mTurk)      | 52.1%        | 47.9%    | 29.5%         | 12.3%         | 6.1%      |
> | CL (expert)     | 57.4%        | 42.6%    | 21.0%         | 12.1%         | 9.5%      |
> | non-CL (expert) | 81.6%        | 18.4%    | 6.4%          | 7.0%          | 5.0%      |
>
> Table 1 shows that CL provides value relative to a random subset of examples which addresses the main feedback above. We see that CL-flagged examples have more of every kind of label issue than non-CL-flagged examples. In our model capacity arguments, we only consider correctable errors. Table 2 shows that using non-CL-flagged examples (6.4% correctable) versus CL-identified examples (>20% correctable) yields significantly fewer correctable examples needed to achieve our destabilization and model capacity findings, further supporting the need for our CL pre-filtering approach to more efficiently discover label errors under human verification budget constraints.
>
> To link this result to the mTurk results in the paper, we further show in Table 2 that the findings for CL-flagged examples are similar for both our expert and mTurk validation experiments on the entire set of CL-flagged label errors.

---

> > ### Author Response · Authors · 2021-07-15
> > **Continued answer from above**
> >
> >
> > Motivated by these preliminary findings during the time-limited rebuttal phase, we will extend this validation experiment for the next revision to include at least 2 examples (1 CL-identified and 1 non-CL identified) for each of the 1000 classes in ImageNet. Each example will be reviewed by at least two expert raters. When disagreement occurs, the two raters will discuss the example and come to consensus.
> >
> >
> > > Only a subset of the label errors had been corrected by humans. It is unclear how many label errors are missed. A complete set of corrected labels should include all label errors.
> >
> > It would have been nice to validate every example, but several of the datasets contain over one million examples, and our budget is limited. Confident learning mitigates this cost with pre-selection, as you mentioned in the strengths of this work. For all but the two largest (and therefore most expensive to validate) datasets, we verify using Mechanical Turk every putative label error found by confident learning. To study the fraction of errors in the data that was not selected by CL as potentially erroneous, we did a small-scale expert validation experiment, detailed below.
> >
> > > It would be helpful if the author can provide more intuitions on why a larger model performs worse in the corrected label set.
> >
> > A larger model performs worse in the corrected label set because it fits the systematic label noise present in the original (train + test) dataset, and once those systematic errors are corrected in the test set, the model performs worse on that data. We have some discussion about this in the last paragraph of section 6; we leave a rigorous analysis of this phenomenon to future work.
> >
> > > Other label error detection methods may also be helpful.
> >
> > We used confident learning because it was recently shown to significantly outperform seven recent high-performing methods for learning with noisy labels and finding label errors (Northcutt, Jiang, & Chuang, 2021) on similar datasets to those we evaluate in this work (e.g., CIFAR-10 and ImageNet). Thank you for bringing this missing explanation to our attention. We will include a note about this in the next revision of the paper.

---

### Decision · Program_Chairs · 2021-07-26

**Decision:**

Accept

**Comment:**

This dataset concerns label errors in test sets for ML benchmarks; the reviewers found it well-documented, motivated, and overall a compelling contribution.